# An Image Encryption Scheme Combining 2D Cascaded Logistic Map and Permutation-Substitution Operations

De Rosal Ignatius Moses Setiadi *[ID] and Nova Rijati

Faculty of Computer Science, Dian Nuswantoro University, Semarang 50131, Indonesia; nova.rijati@dsn.dinus.ac.id
* Correspondence: moses@dsn.dinus.ac.id

**Abstract:** Confusion, diffusion, and encryption keys affect the quality of image encryption. This research proposes combining bit- and pixel-level permutation and substitution methods based on three advanced chaotic logistic map methods. The three chaotic methods are the 2D Logistic-adjusted-Sine map (2D-LASM), the 2D Logistic-sine-coupling map (2D-LSCM), and the 2D Logistic ICMIC cascade map (2D-LICM). The encryption method's design consists of six stages of encryption, involving permutation operations based on chaotic order, substitution based on modulus and bitXOR, and hash functions. Hash functions are employed to enhance key space and key sensitivity quality. Several testing tools are utilized to assess encryption performance, including histogram and chi-square analysis, information entropy, correlation of adjacent pixels, differential analysis, key sensitivity and key space analysis, data loss and noise attacks, NIST randomness tests, and TestU01. Compared to using a single 2D logistic map, the amalgamation of bit-level and pixel-level encryption and the utilization of three 2D cascade logistic maps has improved encryption security performance. This method successfully passes the NIST, TestU01, and chi-square tests. Furthermore, it outperforms the previous method regarding correlation, information entropy, NPCR, and UACI tests.

**Keywords:** chaotic image encryption; enhanced confusion-diffusion; image cryptography; mixed chaotic maps; multi-level image encryption





## 1. Introduction

Image encryption is the process of securing an image by converting the original image into an unreadable form. Without an appropriate decryption key, the image cannot be restored and understood [1,2]. Image encryption aims to protect the privacy and security of images, especially in situations where images are sensitive and confidential, such as medical [3,4], and military [5] images. In the context of encryption methods, Shannon's theory emphasizes the importance of the two main concepts of confusion and diffusion [3,6,7]. Combining these two concepts aims to create a strong dependency between encrypted data and encryption keys. Confusion in the encryption process makes the statistical relationship between the original and encrypted images complex and challenging to predict, while diffusion distributes information in its original form evenly and efficiently to all parts of the encrypted image. Both confusion and diffusion can be achieved in the encryption method by performing intensive permutations and substitutions for each pixel and/or image bit. Permutation involves randomizing the order of bits or pixels in an image [8], aiming to hide the statistical relationship between pixels in an image. Substitution is carried out by changing the bit or pixel value in the image with a different value. Substitution can be carried out using various methods such as substitution-box (S-box) [9–12], XOR [13,14], and modulus [15,16] operations. With increasingly chaotic patterns of permutations and substitutions, it will be more complex and difficult to predict. Therefore, a combination of permutations and substitutions must be performed at the bit- and pixel level.

The important thing to note in the encryption method is the key. Keys play a critical role in the security and confidentiality of encrypted data. Therefore, the key must fulfil several requirements, such as uniqueness, randomness, length or space, and complexity [17–19]. This is so that the input key from the user to encrypt the image needs to be processed first to generate a more complex key, such as a Pseudorandom Number Generator (PRNG) [5,20] and hash function [21]. However, these keys can also be compromised with chosen plaintext attacks if only standard operations are performed, such as stream image shuffling (permutation), image blocking, and sub-image encryption [22]. More complex and combined operations can certainly improve security. In today's modern image encryption, keys, including the chaotic method, can be made very complex. In the key terminology, the chaotic method is used to produce chaotic sequences. Chaotic sequences are susceptible to initial conditions, with different initial parameters resulting in entirely different chaotic sequences, thus providing a very high level of security [18,19,23,24]. The chaotic method has many variations, some famous for image encryption, such as the logistic map, Henon map, Tent map, Arnold map, and Lorentz system [6]. Others include memristive hyperchaotic maps [25] and their development for 3D settings [26]. Focus on the logistic map method. This method has been developed and has several derivatives such as an Improved Logistic Map (ILM) [7], 2D Logistic-adjusted-Sine map (2D-LASM) [27], 2D Logistic-sine-coupling map (2D-LSCM) [28], and 2D Logistic ICMIC cascade map (2D-LICM) [29]. Derivatives from the logistic map have several advantages, namely an increase in the Lyapunov exponent (LE), which enhances randomness and sensitivity, resulting in a more chaotic, unique, and secure sequence. Of the four logistic map derivatives, 2D-LASM, 2D-LICM, and 2D-LSCM have the advantage of having more dimensions and higher complexity. Based on the literature, this study has the following objectives and contributions:

1. Proposing a combination of 2D-LASM, 2D-LICM, and 2D-LSCM to improve image encryption security based on various assessments.
2. Proposing a combination of substitution and permutation techniques based on chaotic sequences at the bit and pixel levels in six stages to improve confusion and diffusion quality.
3. Using a hash function on the private key to produce greater key space before using it to generate a chaotic sequence.

The rest of this paper is written in several sections, namely: related work, which goes into more detail about the hypotheses and work related to this research; a section outlining the proposed approach, which explains the stages in the proposed method; the results and discussion section, which presents research results, discussions, and comparisons with related methods; and lastly is the conclusion, which presents conclusions and research suggestions.

## 2. Related Work

The previous sections have discussed the importance of confusion and diffusion qualities in image encryption, which are implemented through permutation and substitution processes. Another aspect to note is the quality of the encryption keys. Literature related to the development of permutation and substitution techniques and the improvement of key qualities has motivated this research. Initially, permutation and/or substitution encryption methods were applied only to pixels. Furthermore, permutations and substitutions at the bit level have become a trend, as seen in studies [27,29–33]. Bit-level encryption can be carried out on the bit plane or by directly converting the image into bits for the encryption process. One simple method is proposed by [33], which involves the conversion of the RGB image into a bit form. First, the image channels are split, then each channel is converted into a bit form and integrated. After integration, permutation-diffusion is carried out using the chaotic tent map method. Study [31] also employs a fairly simple bit-level technique, namely the bit swapping method and the modulus operation with the piecewise linear chaotic map (PWLCM).

Study [30] employs a more complex algorithm on grayscale images. The image matrix is converted using the bit-plane decomposition technique and then transformed into a vector form. Next, the diffusion process is carried out with cyclic shift permutation and modulus substitution functions for confusion. The diffusion and confusion stages are conducted in several rounds. The results are transformed into a bit-plane form and finally restored into an encrypted image matrix. Further research [29] proposed an improved method, with the main contribution being the use of 2D-LICM, which possesses a much higher Lyapunov exponent (LE) value than the standard logistic map. Encryption is accomplished by converting the image into bit-plane decomposition, followed by cyclic shift and XOR operations on rows and columns based on 2D-LICM. Furthermore, the encrypted bits are reassembled into an encrypted image matrix. Apart from 2D-LICM, other methods have also been developed, such as 2D-LASM [27], whose LE is also better than the standard logistic map. A study [32] introduces a new combined chaotic system (NCCS) based on three chaotic models, namely the Logistic-Sine map, Logistic-Tent map, and Sine-Tent map. These three chaotic models are combined with a hash function to generate the keystream. During the encryption stage, two levels of bit confusion and one label bit diffusion are carried out.

Pixel-level image encryption is proposed once again by [13]. Although it appears simple, the advantage of this method lies in its potential implementation in future quantum computing technology. Hilbert scrambling and XOR operations are performed to encrypt the image. The combination of bit- and pixel-level encryption is proposed in the studies [28,34–36]. Study [34] proposes four levels of encryption. The first involves pixel-level permutation, the second is a new permutation, the third is column permutation, and finally, the fourth is the bitXOR-based diffusion block. These four levels utilize the same chaotic method, namely PWLCM. Another study [28] suggests 2D-LSCM for performing two levels of image encryption. The first level is based on scrambling using 2D-LSCM on rows and columns of image pixels. The second level also employs 2D-LSCM on the rows and columns of image bits. Study [35] introduces the Logistic-Chebyshev map (LCM) and employs the SHA-512 hash operation to enrich the keys pace. The encryption operations conducted in this study include row and column scrambling on pixels, circular shift on bit-planes, and XOR diffusion. Study [36] proposes image encryption by combining the Hilbert curve, cyclic shift, and 2D Henon map. The Hilbert curve is utilized for pixel permutations, cyclic shifts are employed for bit permutations, and 2D Henon maps are used for diffusion processes.

Based on the reviewed research above, it is apparent that the proposed encryption methods are generally developed using bit-level encryption techniques with reliable chaotic methods such as [27,29–33]. Furthermore, a combination of bit- and pixel-level encryption was developed [28,34–36]. We conclude that chaotics such as 2D-LASM, 2D-LICM, and 2D-LSCM have the potential to achieve better encryption quality if combined at both the bit- and pixel levels. Additionally, the use of the hash function will increase the key space. Thus, 2D-LASM, 2D-LICM, 2D-LSCM, and the hash operation are proposed in this study. A more detailed description of the proposed method is presented in Section 3.

## 3. Proposed Approach

The approach suggested in this research involves integrating three distinct types of chaotic maps, namely 2D-LASM, 2D-LSCM, and 2D-LICM. These chaotic methods are utilized in the permutation and substitution processes, operating at both the bit- and pixel levels across six encryption stages. Figure 1 illustrates the proposed method, while the specific details of the method are outlined as follows:

1.  Read the user key and plain image as input for the SHA-512 hash function, and the output is 64 characters each, i.e., *hash A* and *hash B*.

2. Generate initial value $(x0, y0)$ for 2D-LSCM, $(x1, y1)$ for 2D-LASM, and $(x2, y2)$ for 2D-LICM. To generate these, use Equations (1)–(6).

$$x0 = \sum_{n=0}^{\infty} \frac{\sigma(hashA(hA_1, \ldots, hA_{16}))}{10^n}, \text{ subject to } x0 > 1 \tag{1}$$

$$y0 = \sum_{n=0}^{\infty} \frac{\sigma(hashA(hA_{17}, \ldots, hA_{32}))}{10^n}, \text{ subject to } y0 > 1 \tag{2}$$

$$x1 = \sum_{n=0}^{\infty} \frac{\sigma(hashA(hA_{33}, \ldots, hA_{48}))}{10^n}, \text{ subject to } x1 > 1 \tag{3}$$

$$y1 = \sum_{n=0}^{\infty} \frac{\sigma(hashA(hA_{49}, \ldots, hA_{64}))}{10^n}, \text{ subject to } y1 > 1 \tag{4}$$

$$x2 = \sum_{n=0}^{\infty} \frac{\sigma(hashB(hB_1, \ldots, hB_{32}))}{10^n}, \text{ subject to } x2 > 1 \tag{5}$$

$$y2 = \sum_{n=0}^{\infty} \frac{\sigma(hashB(hB_{33}, \ldots, hB_{64}))}{10^n}, \text{ subject to } y2 > 1 \tag{6}$$

3. Generate chaotic sequence-based 2D-LSCM using Equation (7).

$$\begin{cases} x_{i+1} = \sin(\pi(4\gamma x_i(1 - x_i) + (1 - \gamma)\sin(\pi y_i))) \\ y_{i+1} = \sin(\pi(4\gamma y_i(1 - y_i) + (1 - \gamma)\sin(\pi x_{i+1}))) \end{cases} \tag{7}$$

where $\gamma \in [0, 1]$, $x_i = x0$, and $y_i = y0$.

4. Generate chaotic sequence-based 2D-LASM using Equation (8).

$$\begin{cases} x_{i+1} = \sin(\pi\mu(y_i + 3)x_i(1 - x_i)) \\ y_{i+1} = \sin(\pi\mu(x_{i+1} + 3)y_i(1 - y_i)) \end{cases} \tag{8}$$

where $\mu \in [0, 1]$, $x_i = x1$, and $y_i = y1$.

5. Generate chaotic sequence-based 2D-LICM using Equation (9).

$$\begin{cases} x_{i+1} = \sin\left(\frac{21}{(\alpha(y_i+3)\beta x_i(1-\beta x_i))}\right) \\ y_{i+1} = \sin\left(\frac{21}{(\alpha(\beta x_i+3)y_i(1-y_i))}\right) \end{cases} \tag{9}$$

where $\alpha \in [0, +\infty]$, $\beta \in [0, +\infty]$, $x_i = x2$, and $y_i = y2$.

6. Each 2D-LSCM, 2D-LASM, and 2D-LICM has two sequences. Sort the first sequence of 2D-LSCM. Based on the sorting index, perform pixel permutation to perform first-stage encryption.

7. In the second stage of encryption, first transform the second sequence 2D-LSCM with Equation (10). Next, perform bitXOR substitution between the encrypted first stage image and the transformed second sequence of 2D-LSCM.

$$s1 = mod\left(\left(\alpha_1 \times 10^{10}, \ldots, \alpha_n \times 10^{10}\right), 256\right) \tag{10}$$

$$enc2 = (s1_1 \oplus e1_1, \ldots, s1_n \oplus e1_n)$$

where second sequence of 2D-LSCM $(s1) \in (\alpha_1, \ldots, \alpha_n)$, $enc1 \in (e1_1, \ldots, e1_n)$, $\oplus$ is a symbol of the bitXOR operator.

8. In the third stage, encryption is conducted using bit-level permutation, so the encrypted second-stage image ($enc2$) needs to be converted to binary form. At this

stage, sort the first 2D-LASM sequence. Then, perform permutation *enc2* based on the sorting index.

9. The image, which is still in binary form, is restored to decimal form (*enc3*) to perform pixel substitution in the fourth stage. At this stage, the second sequence of 2D-LASM must be converted with Equation (11), then carry out the modulus operation with Equation (12).

$$s2 = mod\left(\left(\beta_1 \times 10^{10}, \ldots, \beta_n \times 10^{10}\right), 256\right) \tag{11}$$

$$enc4 = mod((s2_1 - e3_1, \ldots, s2_n - e3_n), 256) \tag{12}$$

where second sequence of 2D-LASM (*s2*) $\in (\beta_1, \ldots, \beta_n)$, *enc3* $\in (e3_1, \ldots, e3_n)$.

10. The bit-level permutation is performed in the fifth stage based on the first 2D-LICM sequence. Then, the image is converted back into binary form, then sorting the first 2D-LICM sequence. Then, perform a permutation based on the first 2D-LICM index sorting sequence in the encrypted fourth-stage image (*enc4*).

11. Restore the encrypted fifth stage image to decimal form, then convert the second sequence from 2D-LICM with Equation (13). Then, perform the bitXOR operation on the encrypted fifth stage image (*enc5*) with the second sequence from 2D-LICM to obtain the final encrypted image (*fe*), see Equation (14).

$$s3 = mod\left(\left(\gamma_1 \times 10^5, \ldots, \gamma_n \times 10^5\right), 256\right) \tag{13}$$

$$fe = (s3_1 \oplus e5_1, \ldots, s3_n \oplus e5_n) \tag{14}$$

where second sequence of 2D-LICM (*s3*) $\in (\gamma_1, \ldots, \gamma_n)$, *enc5* $\in (e5_1, \ldots, e5_n)$.

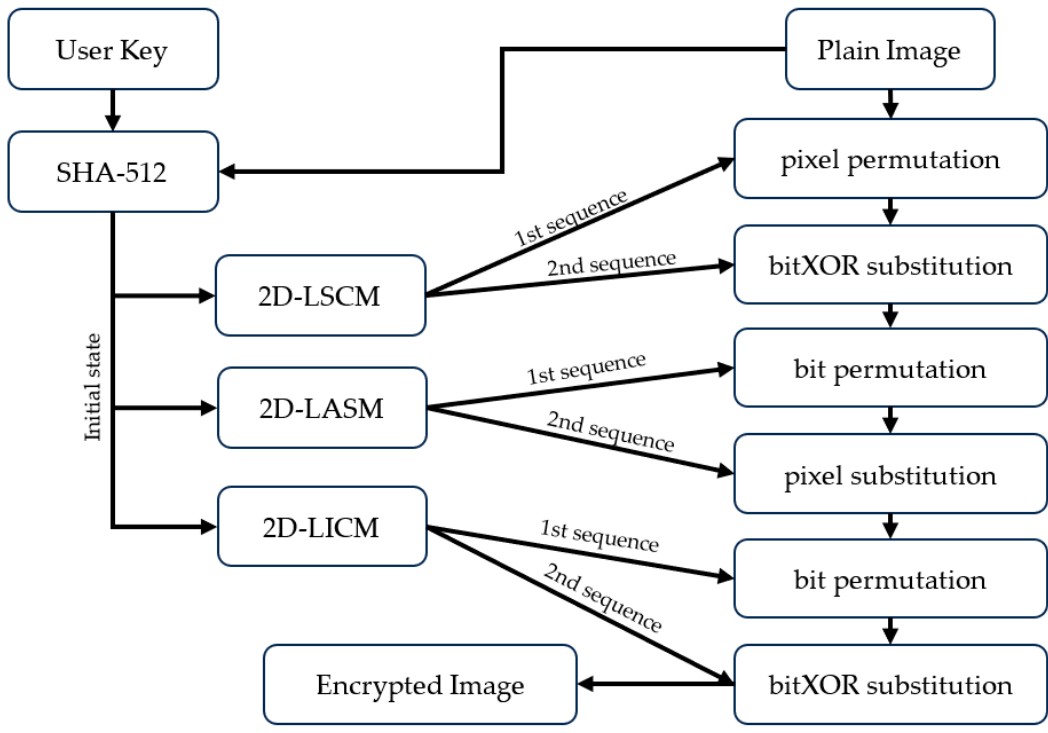

**Figure 1.** Illustration of proposed encryption method.

The encryption method proposed above has six stages consisting of one pixel-level permutation stage, two bit-level permutation stages, a one-pixel substitution stage, and two bitXOR substitution stages. Meanwhile, perform image decryption can be performed with the reverse step.

## 4. Results and Discussion

The experiments conducted in this research were performed using Matlab R2021a, an i7 11th gen processor with 16 GB of memory. Two types of images were tested, namely 8-bit grayscale and 24-bit color (red, green, and blue) images. The images used were standard test images and random samples from the BossBase dataset. The image dimensions were $512 \times 512$ pixels, and a sample image used is presented in Figure 2. It is important to note that the proposed method is primarily designed for 8-bit images. The channels are separated first for color images, and then the proposed method is applied to each channel. After encryption, all channels are combined into one encrypted color image. Sample encrypted results for grayscale images are shown in Figure 3, while Figure 4 displays the results for color images. To assess image encryption security, several tests, such as histogram and chi-square ($\chi^2$), entropy, correlation coefficient, differential, key space, and key sensitivity analysis. The NIST and TestU01 randomness suite tests, noise and data loss attacks, and ablation studies are conducted, as presented in Sections 4.1–4.10.

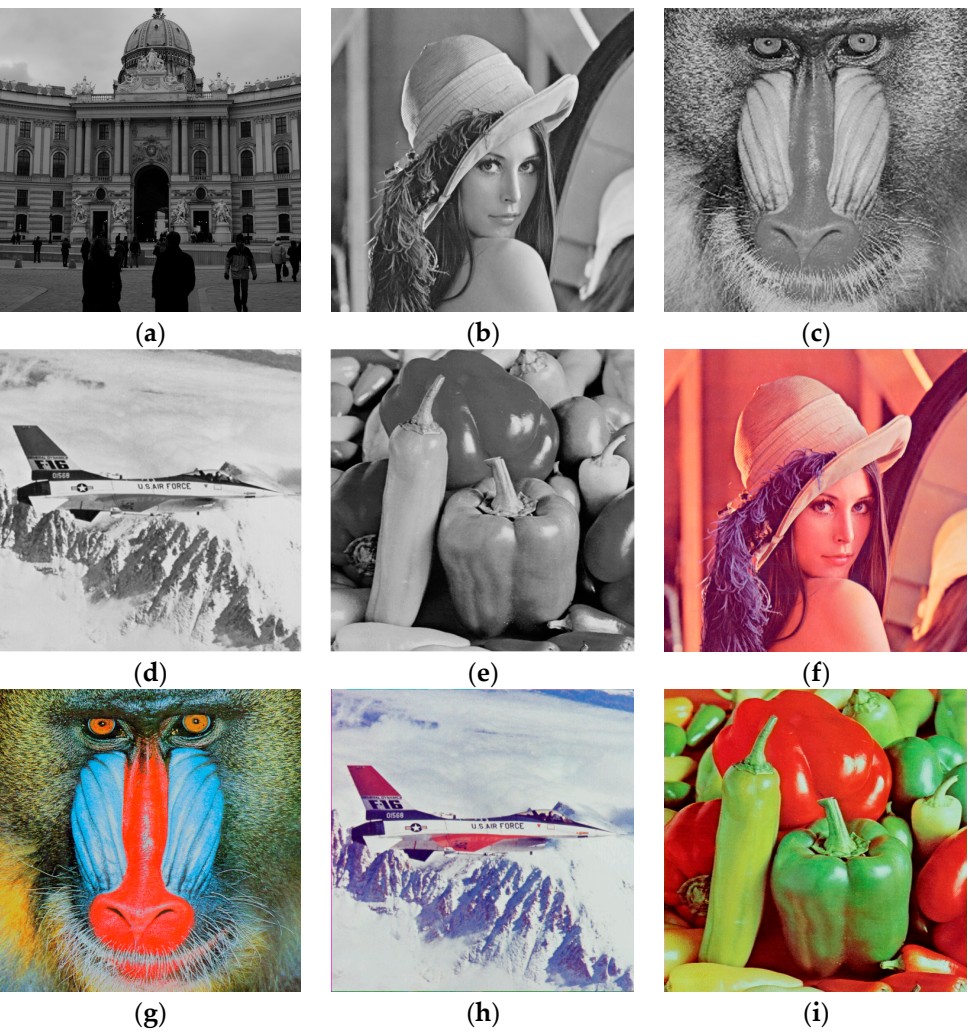

**Figure 2.** Sample image dataset. (**a**) 1013.pgm; (**b**) Lena grayscale; (**c**) baboon grayscale; (**d**) airplane grayscale; (**e**) peppers grayscale; (**f**) Lena color; (**g**) baboon color; (**h**) airplane color; and (**i**) peppers color.

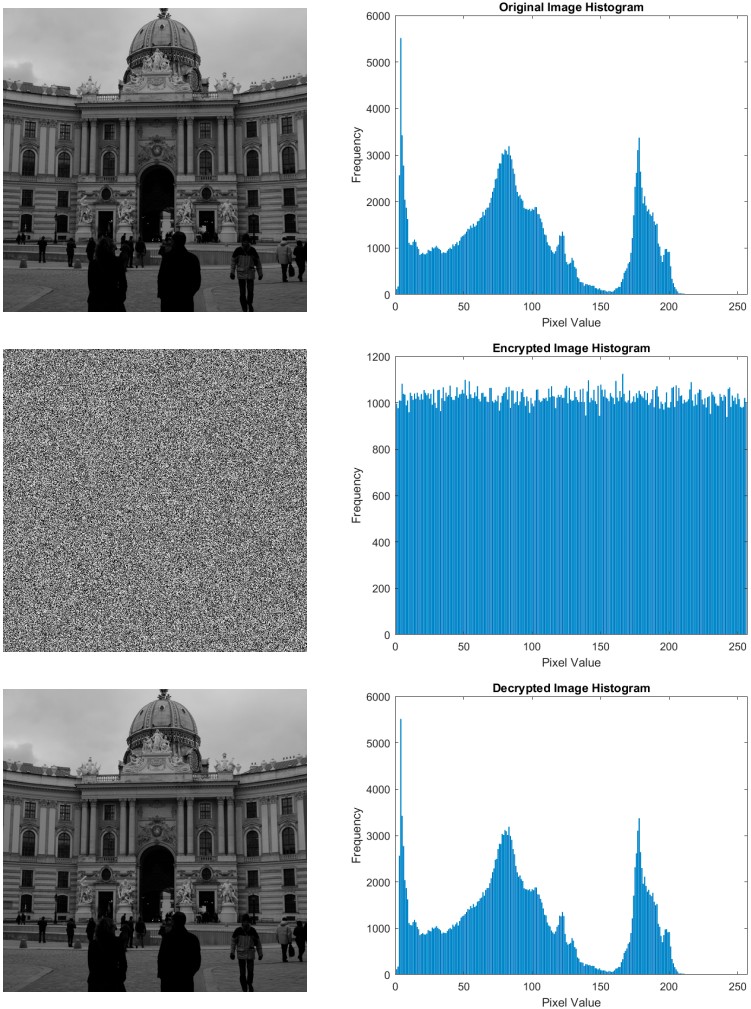

**Figure 3.** Sample results of grayscale image (row 1: original image; row 2: encrypted image; row 3: decrypted image).

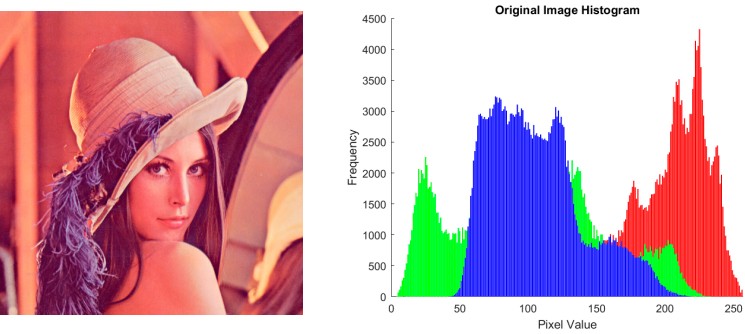

**Figure 4.** *Cont.*

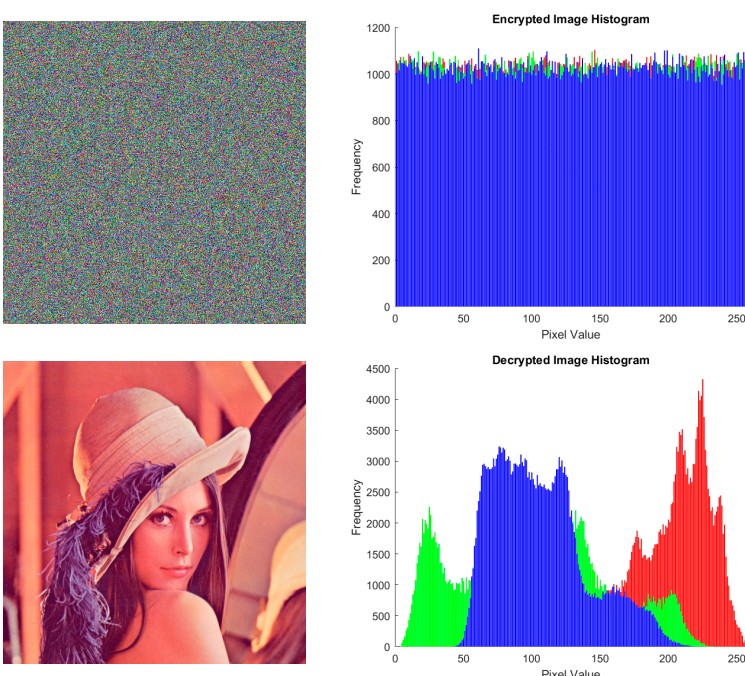

**Figure 4.** Sample results of the color image (row 1: original image; row 2; encrypted image; row 3: decrypted image).

### 4.1. Histogram and Chi-Square ($X^2$) Test

The image histogram is a visual representation of the pixel intensity distribution in the image. Histograms can provide important information about image characteristics, including contrast, brightness, intensity distribution, and color variation. In the context of image encryption, the image histogram needs to change significantly. Besides that, the histogram distribution must be nearly uniform. The histogram of the encrypted image presented in Figures 3 and 4, row 2, appears visually uniform. However, validation of histogram uniformity was carried out using $X^2$ analysis. If the calculated $\chi^2$ value is less than or equal to $X^2_{\delta,fd} = 293.2478$ with a significance level ($\delta$) of 0.05 and freedom degrees ($fd$) of 255, it indicates that the histogram is considered to be uniformly distributed. Equation (15) is used to calculate the chi-square value.

$$X^2 = \sum_{i=1}^{256} \frac{(r_i - r/255)^2}{r/255} \tag{15}$$

In Matlab, the index range for *i* is from 1 to 256, as it starts indexing from 1. The grey recurrence value ($r_i$) represents the value assigned to each occurrence of the *i*th grey value.

Based on the results presented in Table 1, all image histograms are confirmed to be uniform. This is evidenced by the $X^2$ value, which is smaller than 293.2478. This also proves that the performance of the proposed encryption method is excellent based on the histogram and chi-square analysis. As a note, for RGB images, the chi-square value is taken from the mean value of the R, G, and B channels.

**Table 1.** Chi-square test results.

| Image | Chi-Square (Mean) | Passed? |
|:---:|:---:|:---:|
| 1013.pgm | 275.9563 | pass |
| Lena grayscale | 212.6831 | pass |
| Baboon grayscale | 284.3348 | pass |
| Airplane grayscale | 285.0325 | pass |

**Table 1.** *Cont.*

| Image | Chi-Square (Mean) | Passed? |
|---|---|---|
| Peppers grayscale | 259.1784 | pass |
| Lena color | 209.9740 | pass |
| Baboon color | 226.6225 | pass |
| Airplane color | 251.3143 | pass |
| Peppers color | 289.0926 | pass |
| Average | 254.9098 | pass |

*4.2. Correlation Coefficient of Adjacent Pixel Test*

Analyzing the correlation coefficient (*cor*) between adjacent pixels is a technique employed to assess the level of interdependence between neighboring pixels in an image following the encryption process. This analysis evaluates the level of randomness or irregularity in the arrangement of encrypted pixels. In addition, this analysis can indicate patterns or structures that still exist in encrypted images. The range of *cor* measurements is −1 to 1. A value closer to −1 or 1 shows an inverse or high correlation, while the optimal value is close to zero. A value closer to zero indicates the minimum correlation, signifying encryption at the highest level of randomness. Equation (16) is used to calculate the *cor*.

$$cor_{a,b} = \frac{\frac{1}{N}\sum_{i=1}^{N}[a_i - E(a)][b_i - E(b)]}{\sqrt{\frac{1}{N}\sum_{i=1}^{N}[a_i - E(a)]^2}\sqrt{\frac{1}{N}\sum_{i=1}^{N}[b_i - E(b)]^2}} \tag{16}$$

Equation (16) uses the symbol $N$ to represent the total number of pixels in the image. The variables $a$ and $b$ refer to two neighboring pixels positioned diagonally, horizontally, or vertically. $E(a)$ and $E(b)$ represent the expectations or average values of $a$ and $b$, respectively. Figure 5 represents a sample of the correlation coefficient of adjacent pixels plot results.

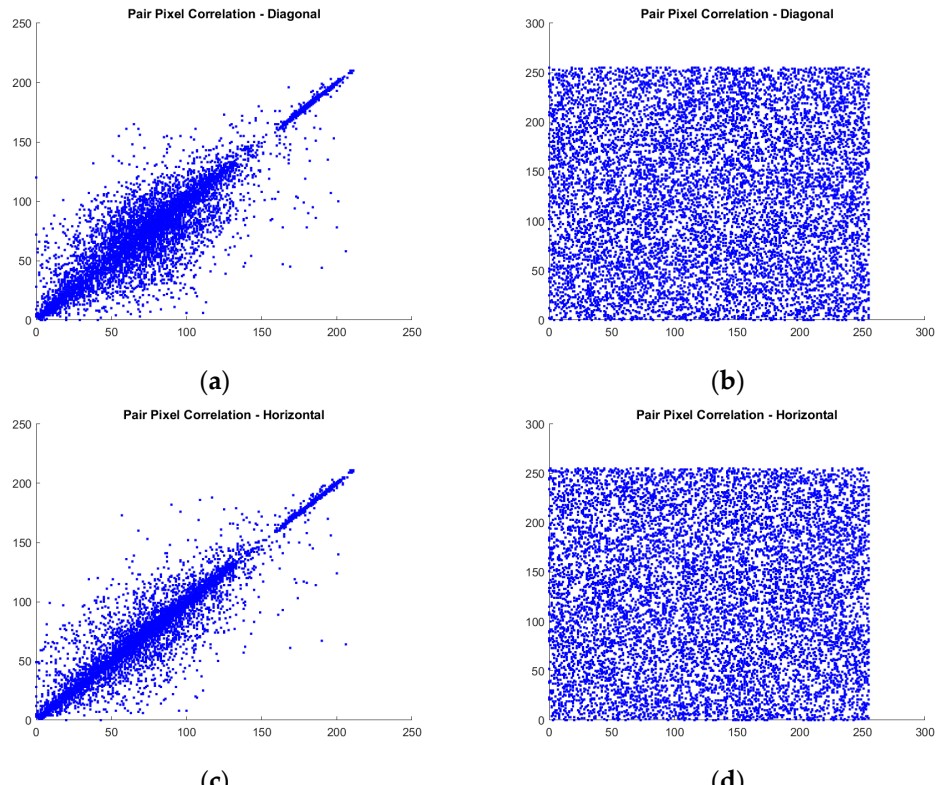

**Figure 5.** *Cont.*

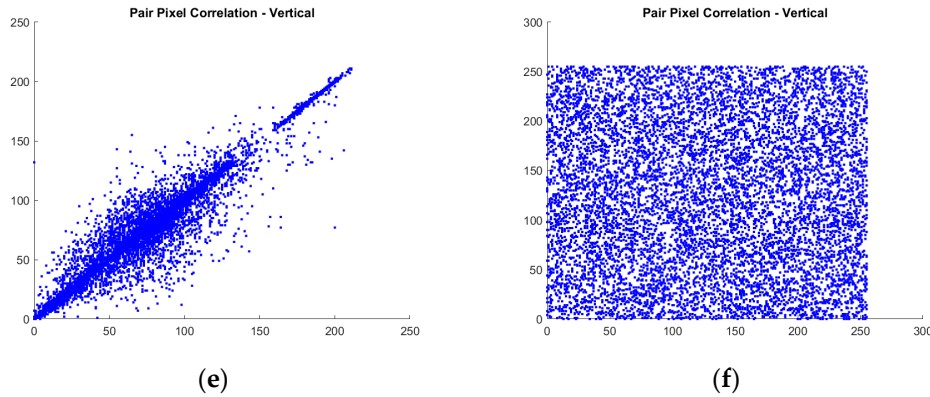

**Figure 5.** Sample results of plot correlation coefficient of adjacent pixels of 1013.pgm image: (**a**) diagonal of the original image; (**b**) diagonal of encrypted image; (**c**) horizontal of the original image; (**d**) horizontal of encrypted image; (**e**) vertical of the original image; (**f**) vertical of encrypted image.

The results presented in Table 2 show that all images correlate closely to zero. This indicates that the proposed method performs well based on the analysis. In Table 3, a correlation comparison with related work is also carried out on the Lena image, and it can be seen that the proposed method has the best correlation value in the vertical direction because it is closest to zero. However, compared to related methods, the proposed method is superior in diagonal correlation (marked with bold text), while horizontal and vertical correlation is the second best (marked with underlined text). It can also be seen that no related method excels at all correlation values, while the proposed method holds a relative advantage across all correlation values. As a note, the correlation value is taken from the average value of the R, G, and B channels for RGB images.

**Table 2.** The correlation coefficient of adjacent pixel results of the encrypted image.

| Image | Horizontal | Vertical | Diagonal |
|---|---|---|---|
| 1013.pgm | 0.0013 | 0.0018 | −0.0003 |
| Lena grayscale | −0.0011 | 0.0005 | 0.0007 |
| Baboon grayscale | 0.0016 | 0.0019 | 0.0017 |
| Airplane grayscale | −0.0015 | −0.0014 | 0.0012 |
| Peppers grayscale | 0.0017 | 0.0019 | 0.0018 |
| Lena color | 0.0005 | 0.0018 | 0.0006 |
| Baboon color | −0.0016 | −0.0001 | −0.0019 |
| Airplane color | −0.0009 | 0.0012 | 0.0014 |
| Peppers color | 0.0002 | −0.0014 | 0.0017 |

**Table 3.** Comparison of correlation coefficient of the adjacent pixel of encrypted Lena Image (grayscale).

| Method | Horizontal | Vertical | Diagonal |
|---|---|---|---|
| Ref. [27] | 0.0013 | 0.0006 | 0.0019 |
| Ref. [29] | 0.0019 | 0.0012 | <u>0.0009</u> |
| Ref. [32] | **0.0005** | −0.0025 | 0.0028 |
| Ref. [35] | 0.0035 | −0.0011 | −0.0028 |
| Ref. [36] | −0.0016 | **0.0003** | −0.0022 |
| Proposed | <u>−0.0011</u> | <u>0.0005</u> | **0.0007** |

### 4.3. Information Entropy Test

Entropy measures the level of randomness or uncertainty in data distribution. In image encryption terminology, entropy can be used to evaluate the effectiveness of encryption methods to measure the degree of randomness or uncertainty in data distribution. The

entropy of 8-bit images generally has a range of 0 to 8, where high entropy means an increasingly random distribution of pixel values, making it difficult to guess the actual pixel value. Conversely, low entropy indicates a clear pattern or dependency between pixel values, making the image vulnerable to statistical attacks. Entropy can be measured by Equation (17).

$$H = \sum_{i=1}^{n} p(e_i) log_2 \left( \frac{1}{p(e_i)} \right) \tag{17}$$

where $H$ is entropy, which is calculated to involve the total number of symbols ($n$), the information of the source (encrypted image) represented by $e_i$, and the probability of occurrence of the $e_i$, represented by $p(e_i)$.

Based on the results presented in Table 4, all images have very high entropy values: the lowest is 7.9993 and the highest is 7.9994. It can also be concluded that the entropy value is very stable, both in grayscale and RGB images. In RGB images, the entropy value presented is the mean value of all channels. The results presented in Table 5 also confirm that the proposed method has an advantage in entropy values compared to related methods.

**Table 4.** Information entropy results.

| Image | Information Entropy |
|---|---|
| 1013.pgm | 7.9994 |
| Lena grayscale | 7.9994 |
| Baboon grayscale | 7.9993 |
| Airplane grayscale | 7.9993 |
| Peppers grayscale | 7.9994 |
| Lena color | 7.9993 |
| Baboon color | 7.9994 |
| Airplane color | 7.9993 |
| Peppers color | 7.9993 |
| Average | 7.9993 |

**Table 5.** Comparison of information entropy of encrypted Lena image.

| Method | Information Entropy |
|---|---|
| Ref. [29] | 7.9973 |
| Ref. [32] | **7.9994** |
| Ref. [35] | 7.9973 |
| Ref. [36] | <u>7.9987</u> |
| Proposed | **7.9994** |

### 4.4. Key Sensitivity Test

The key sensitivity test is essential in image encryption, and its purpose is to evaluate the sensitivity of the encryption to small changes in the encryption key. During image encryption, a transformation occurs in the original image, which the encryption key affects. By introducing variations to the encryption key, key sensitivity tests examine the impact of these changes on the resulting encrypted image. To test key sensitivity, at least two encryptions are performed on an image with different keys. This difference is generally a single bit, which can be at the key's start, end, or middle. The sample test results presented in Figure 6 confirm that the proposed method satisfies the requirements of this test, as even a single-bit difference results in significant variation, making precise image decryption impossible.

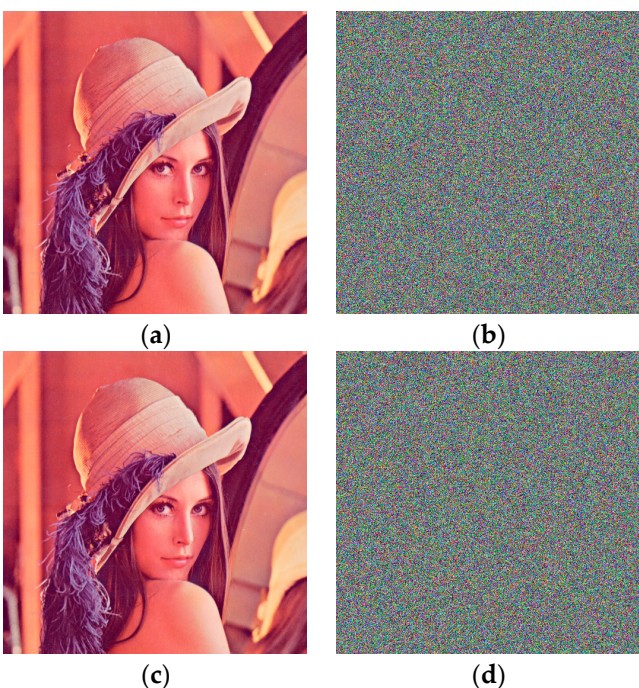

**Figure 6.** Sample results of key sensitivity decryption test: (**a**) original image; (**b**) encrypted image; (**c**) decrypted image with correct key; (**d**) decrypted image with single-bit key modification.

### 4.5. Differential Test

The differential test in image encryption is a crucial evaluation process commonly employing the Normalized Pixel Change Rate (NPCR) and Unified Average Changing Intensity (UACI) metrics. NPCR assesses the impact of changes in the encryption key on the resulting encrypted image by calculating the percentage of differing pixels between two encrypted images generated using slightly different encryption keys. A higher NPCR value indicates that even slight changes in the encryption key lead to significant variations in the encrypted image. On the other hand, UACI measures the average intensity changes in encrypted images caused by modifications in the encryption key. UACI quantifies the difference in pixel intensity between two encrypted images produced using slightly different encryption keys. A higher UACI value signifies those alterations in the encryption key result in notable intensity changes in the encrypted image. The optimal NPCR value is approximately 99.6094%, while the ideal UACI value is around 33.4635%. Equations (18) and (19) are utilized to calculate NPCR and UACI, respectively.

$$
\begin{aligned}
NPCR &= \left[ \frac{1}{N \times M} \sum_{i=1}^{N} \sum_{j=1}^{M} Diff(i,j) \right], \\
Diff(i,j) &\begin{cases} 0 \ if \ C1(i,j) = C2(i,j) \\ 1 \ if \ C1(i,j) \neq C2(i,j) \end{cases}
\end{aligned}
\tag{18}
$$

$$
UACI = \left[ \frac{1}{N \times M} \sum_{i=1}^{N} \sum_{j=1}^{M} \frac{|C1(i,j) - C2(i,j)|}{255} \right]
\tag{19}
$$

where $C1$ and $C2$ represent the original cipher and the altered cipher, respectively. $N$ and $M$ correspond to the width and height dimensions, respectively, and $i$ and $j$ indicate the coordinates of individual pixels.

Based on the NPCR and UACI tests, the proposed method produces an average NPCR and UACI value that is very close to the ideal value, see Table 6. The average NPCR value is 99.9060, the difference is only 0.0004 from the ideal NPCR value. Meanwhile, the average UACI value is 33.4611, and the difference is only 0.0024. This indicates that the performance

based on NPCR and UACI is very satisfying. Based on Table 7, the NPCR value of the proposed method outperforms the related methods. Meanwhile, UACI is the second best.

**Table 6.** NPCR and UACI results.

| Image | NPCR | UACI |
|---|---|---|
| 1013.pgm | 99.6112 | 33.4610 |
| Lena grayscale | 99.6091 | 33.4598 |
| Baboon grayscale | 99.6161 | 33.4693 |
| Airplane grayscale | 99.5987 | 33.4582 |
| Peppers grayscale | 99.6063 | 33.4626 |
| Lena color | 99.6052 | 33.4561 |
| Baboon color | 99.6125 | 33.4655 |
| Airplane color | 99.6131 | 33.4567 |
| Peppers color | 99.6086 | 33.4612 |
| Average | 99.6090 | 33.4611 |

**Table 7.** Comparison of NPCR and UACI of Lena image.

| Method | NPCR | UACI |
|---|---|---|
| Ideal value | 99.6094 | 33.4635 |
| Ref. [29] | **99.6096** | 33.4574 |
| Ref. [32] | 99.6000 | 33.4800 |
| Ref. [36] | 99.6058 | 33.4421 |
| Proposed | 99.6091 | **33.4598** |

*4.6. NIST Randomness Test*

The National Institute of Standards and Technology (NIST) developed a series of tests that are used to assess the randomness or random nature of a data or bit stream. This randomness test serves as a standard for measuring the security of cryptographic algorithms. The NIST randomness test comprises 15 statistical tests, which include tests such as the frequency test, run test, run-bit test, long-term test, and others. For each test, a p-value is generated ranging from 0 to 1. To assess the effectiveness of encryption and ensure compliance with test standards, each test requires a sequence of at least 106 bits and must generate a p-value greater than 0.01 to pass. In terms of image encryption, the results of image encryption can be directly tested. The encrypted image must first be converted into a binary file, then saved with the .dat extension. This .dat file is input for testing through a series of NIST tests. The test results, including the average $p$-values of all encrypted images, are documented in Table 8. The statistical test results obtained from the NIST tool indicate that the proposed method successfully passed all tests and demonstrated resistance to various attacks. The results presented in Table 8 confirm that the proposed method successfully passed all tests, as evidenced by the average p-value of all tests being greater than 0.49.

**Table 8.** Average NIST randomness test results from all images.

| Test Name | *p*-Value | Note |
|---|---|---|
| Frequency | 0.638289477 | Passed |
| Block Frequency | 0.785995714 | Passed |
| Cumulative Sums (Forward) | 0.838654538 | Passed |
| Cumulative Sums (Reverse) | 0.521356511 | Passed |
| Runs | 0.206741782 | Passed |
| Longest Run of Ones | 0.214957335 | Passed |
| Rank | 0.298282198 | Passed |
| Discrete Fourier Transform | 0.747352546 | Passed |
| Nonperiodic Template Matchings | 0.295798124 | Passed |
| Overlapping Template Matchings | 0.726999602 | Passed |
| Universal Statistical | 0.287515082 | Passed |
| Approximate Entropy | 0.815533161 | Passed |
| Random Excursions | 0.36948825 | Passed |
| Random Excursions Variant | 0.251379246 | Passed |
| Serial | 0.29333542 | Passed |
| Linear Complexity | 0.574354885 | Passed |
| Average | 0.491627117 | Passed |

### 4.7. TestU01

TestU01 is a statistical testing software suite used to assess the quality and randomness of a random number generator [37]. In this case, it is the second chaotic sequence because TestU01 requires integer input and all three are converted, as explained in Section 3. To test the randomness, we use two battery tests, namely Rabbit and Alphabit. Rabbit is a test suite designed to evaluate a random number generator's correlation between the generated bits. This test consists of 39 different statistical sub-tests to help identify imbalances or patterns in the distribution of bits produced by the generator. Alphabit consists of 17 more general sub-tests involving basic statistical tests and distribution tests to identify abnormalities in the distribution of random numbers. The number of sub-tests in Rabbit and Alphabit is generally used for bitstreams with a length of $2^{24}$ bits [38–40]. The test results presented in Table 9 show that all chaotic sequences have passed all tests and are proven to exhibit randomness that meets the criteria of TestU01.

**Table 9.** TestU01 results of $2^{24}$-bits length bitstream.

| Chaotic Sequence | Test Name | | Note |
|---|---|---|---|
| | Rabbit | Alphabit | |
| 2D-LASM | 39/39 | 17/17 | All test passed |
| 2D-LSCM | 39/39 | 17/17 | All test passed |
| 2D-LICM | 39/39 | 17/17 | All test passed |

### 4.8. Data Loss and Noise Attack Test

Data loss and noise attack testing in image encryption is an important process to evaluate the robustness of image encryption methods against attacks that result in data loss. This helps determine how the image encryption method can maintain confidentiality, overcome damage, and ensure data integrity despite data loss or increased noise. Additionally, it can help reveal potential weaknesses in image encryption methods against data loss or noise attacks. In this section, the proposed method is tested, as presented in Figure 7. While, visually, the attack of data loss can be seen, the addition of noise may not be obvious. The results of the description show that the proposed method is able to restore the image in all forms with scattered noise. This test confirms the statistical measurement tools that have been tested, such as chi-square, correlation of adjacent pixels, entropy, differential analysis, and NIST. This is evidenced by the even distribution of noise in the decryption process.

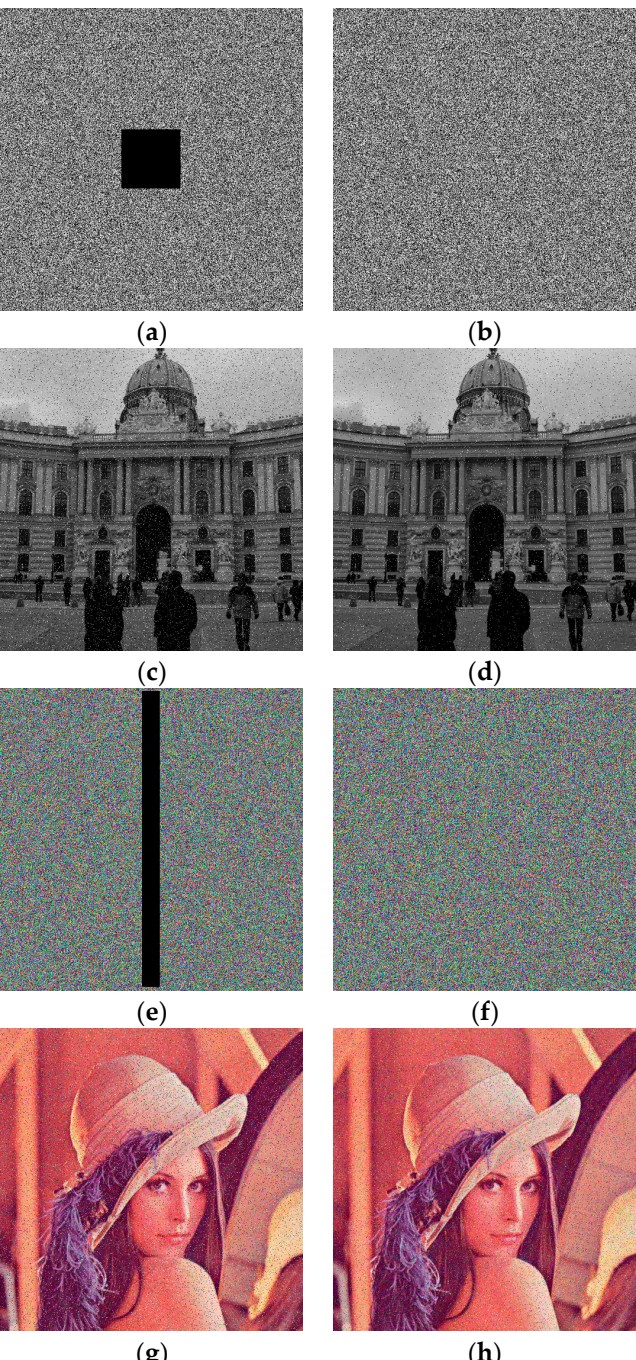

**Figure 7.** Sample results of data loss and noise attack: (**a**) data loss attack of gray image; (**b**) salt and pepper noise attack (0.05) of gray image; (**c**) decrypted gray image of data loss attack; (**d**) decrypted gray image of noise attack; (**e**) data loss attack of color image; (**f**) salt and pepper noise attack (0.05) of color image; (**g**) decrypted color image of data loss attack; (**h**) decrypted color image of noise attack.

*4.9. Key Space Analysis*

The analysis of key space plays a crucial role in image encryption as it encompasses the entire range of potential encryption keys utilized within a specific encryption system. In the context of image encryption, the key space holds significant importance since a larger key space presents a greater challenge in guessing the correct encryption key. The key space must be extensive in an effective encryption system to prevent successful brute force attacks. The key space must have $2^{100}$ or more possibilities [41,42]. The proposed method has several parameters, a dynamic initial value, and a hash operation, from which the

proposed method can calculate the key space. The total key space of the proposed method is $\approx 1.34 \times 10^{154}$, as presented in detail in Table 10. This shows that the proposed method will be highly reliable and resistant to brute-force attacks.

**Table 10.** Total key space of proposed method.

| Method | Key Space | Note |
|---|---|---|
| SHA-512 | $2^{512}$ | - |
| 2D-LASM | $\approx 3 \times 10^{16}$ | Parameter and initial value |
| 2D-LSCM | $\approx 3 \times 10^{16}$ | Parameter and initial value |
| 2D-LICM | $\approx 2 \times 10^{16} + 2 \times 10^{20}$ | Parameter and initial value |
| Total | $\approx 1.34 \times 10^{154}$ | |

*4.10. Ablation Study*

Ablation studies constitute the final part of this section. They are conducted to perform a more in-depth analysis by removing certain components of the proposed encryption method to observe how these changes affect security quality and encryption performance. As explained earlier, the proposed encryption method consists of six stages, with each of the two stages being based on one 2D logistic map. Upon reviewing the results presented in Table 11, one of the weaknesses of the proposed method is its slightly slower encryption time. However, this is reasonable due to combining three 2D logistic maps. Moreover, the encryption time remains reasonable, totaling less than two seconds, with the difference being no more than half a second. Another observation from combining these three 2D logistic maps is the enhancement in all security aspects, as demonstrated by chi-square, IE, CC, NPCR, and UACI assessments. Furthermore, it can be elucidated that the two stages utilizing 2D-LICM have a more significant impact on improving the performance against differential attacks. The two stages employing 2D-LSCM tend to increase IE and CC. Meanwhile, the two stages of 2D-LASM tend to strongly influence the chi-square value. In short, despite the increase in computational time, the proposed method's combination provides notable encryption security advantages.

**Table 11.** Ablation study results.

| Method | Avg $X^2$ | Avg IE | CC 'Lena' | Avg NPCR | Avg UACI | Time Taken (s) |
|---|---|---|---|---|---|---|
| Without 2D-LASM | 285.4343 | 7.9990 | 0.0018 −0.0010 0.0012 | 99.5013 | 33.3942 | 1.253434 |
| Without 2D-LSCM | 274.7353 | 7.9991 | 0.0009 0.0012 −0.0021 | 99.45898 | 33.2167 | 1.214575 |
| Without 2D-LICM | 270.2336 | 7.9973 | −0.0025 0.0016 0.0006 | 99.3849 | 32.9148 | **1.110195** |
| Proposed | **254.9098** | **7.9993** | **−0.0011** **0.0005** **0.0007** | **99.6090** | **33.4611** | 1.584455 |

**5. Conclusions**

In this study, a proposal has been made to combine the 2D-LASM, 2D-LICM, and 2D-LSCM methods for image encryption. Substitution and permutation techniques are also employed based on chaotic sequences at both the bit- and pixel levels. A hash function is also utilized on the private key to enhance the key space quality before generating a garbled sequence. The test results, encompassing histogram and chi-square analysis, information entropy, adjacent pixel correlation, differential analysis, key sensitivity analysis, key space analysis, data loss, noise attacks, NIST randomness test, and TestU01 test, affirm that the

method introduced in this study produced highly satisfactory outcomes. In fact, most of the test results exhibit superior performance when compared to related studies' test results. Based on these findings, it can be concluded that the proposed method significantly enhances image encryption security. The comprehensive test outcomes demonstrate that this method fulfills essential cryptographic requirements, such as randomness, resistance against attacks, and data loss prevention. Hence, this method has the potential to be a strong contender in meeting the demands for privacy protection and image security. This research could be extended in the future by incorporating a memristive concept to enhance computational efficiency and effectiveness.

**Author Contributions:** Conceptualization, D.R.I.M.S.; methodology, D.R.I.M.S.; software, N.R.; validation, D.R.I.M.S. and N.R.; formal analysis, D.R.I.M.S.; investigation, D.R.I.M.S.; resources, N.R.; data curation, N.R.; writing—original draft preparation, D.R.I.M.S.; writing—review and editing, N.R.; visualization, N.R.; supervision, D.R.I.M.S.; project administration, N.R. All authors have read and agreed to the published version of the manuscript.

**Funding:** This research was funded by the Directorate General of Higher Education, Research and Technology of Indonesia, grant number 182/E5/PG.02.00.PL/2023, 026/LL6/PB/AL.04/2023, and 065/A38-04/UDN-09/VII/2023.

**Acknowledgments:** The author is very grateful to LPPM Dian Nuswantoro University and the Ministry of Education, Culture, Research, and Technology of Indonesia for their support for this research.

**Conflicts of Interest:** The authors declare no conflict of interest.

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
