# Peer review of "An Image Encryption Scheme Combining 2D Cascaded Logistic Map and Permutation-Substitution Operations"

_computation, doi:10.3390/computation11090178_

Round 1

Reviewer 1 Report

1.      Change the title to “An Image Encryption Scheme Combining 2D Cascaded Logistic Map and Permutation-substitution Operations”.

2.     Present every equation in a way can be found in any textbook on Calculus or famous monograph. As for the formal presentation of an equation, refer to http://linear.axler.net/LinearAbridged.pdf

3.      Why and how the proposed scheme can solve the security challenges of private-key encryption schemes summarized at https://doi.org/10.1016/j.jisa.2020.102566 should be explained in the introduction part.

4.   Double-check the grammar and Logic flow of every sentence.

NA

Author Response

Reviewer#1, Concern # 1: Change the title to “An Image Encryption Scheme Combining 2D Cascaded Logistic Map and Permutation-substitution Operations”.

Author response:  Thank you for the advice given, we agree.

Author action: We updated the title of manuscript, by marked with a yellow highlight.

Reviewer#1, Concern # 2: Present every equation in a way can be found in any textbook on Calculus or famous monograph. As for the formal presentation of an equation, refer to http://linear.axler.net/LinearAbridged.pdf

Author response:  Thank you for the advice given, we agree.

Author action: We updated section 3 by marking changes with a yellow highlight.

Reviewer#1, Concern # 3:   Why and how the proposed scheme can solve the security challenges of private-key encryption schemes summarized at https://doi.org/10.1016/j.jisa.2020.102566 should be explained in the introduction part.

Author response:  Thank you for the advice given, we agree

Author action: We updated the introduction section by marking changes with a yellow highlight.

Reviewer#1, Concern # 4: Double-check the grammar and Logic flow of every sentence.

Author response:  Thank you for the information, we have double checked our paper and tried our best to minimize grammar and typo errors.

Reviewer 2 Report

Authors present an image encryption method where they combine 2D-LASM, 2D-LSCM, and 2D-LICM chaotic maps to improve the performance at the output. Bit and pixel level permutations are performed and a hash function is employed.

The paper is well organized and well-written. On the other hand, the work is fully based on simulations and the computational costs are not discussed. The authors present their result in Table1 to Table 8 and there are practically no differences between the other existing works. The improvements over the other works should be clarified. The proposed algorithm works well but it is not discussed how the performance of the algorithm will be affected in case of a change in the algorithm steps. For instance, the role of the chaotic maps employed, or the permutation steps, or the hash functions, on the performance are not discussed. In general, the contribution of the work remains unclear to the reader.

Author Response

Reviewer#2, Concern # 1: The paper is well organized and well-written. On the other hand, the work is fully based on simulations and the computational costs are not discussed. The authors present their result in Table1 to Table 8 and there are practically no differences between the other existing works. The improvements over the other works should be clarified. The proposed algorithm works well but it is not discussed how the performance of the algorithm will be affected in case of a change in the algorithm steps. For instance, the role of the chaotic maps employed, or the permutation steps, or the hash functions, on the performance are not discussed. In general, the contribution of the work remains unclear to the reader.

Author response:  Thank you for the advice given, to overcome this we have added section 4.10, hopefully this can make this paper more acceptable.

Reviewer 3 Report

Minor editing of the English language required

Author Response

Reviewer#3, Concern # 1: Why not employing the TestU01 test suite to evaluate the randomness? The authors should give declaration about this point.

Author response:  Thanks for your suggestions, we have added this measurement in section 4.7.

Reviewer#3, Concern # 2: It is well known that the discrete maps can generate chaotic sequences. But they are very sensitive to the initial values. I doubt that how to ensure the generated initial values to trigger chaotic state.

Author response:  Thanks for your comments, we agree on this. In this case we try to determine the best parameters and limit the range of initial values as we have explained in section 3. Even so, we still pay attention to the level of security and as far as the experiments we have carried out, the results prove that the level of security of the proposed method is satisfactory.

Reviewer#3, Concern # 3: Make comparations on encryption performance between the previous works and this work. A table list is better to display this to the readers.

Author response:  Thank you for the advice, we have provided a comparison with previous works as presented in Table 3, Table 5 and Table 7.

Reviewer#3, Concern # 4: Please redrawn the encryption method in Figure 1 by more readable way.

Author response:  Thank you for the advice given, but we think the explanations and illustrations in section 3 have provided a fairly clear picture of the proposed encryption process.

Reviewer#3, Concern # 5: Rich the abstract with main results

Author response:  Thank you for your advice, we updated the abstract part marked with turquoise highlight.

Reviewer#3, Concern # 6: How to display the bit-pixel level permutation-substitution?

Author response:  Thanks for your comment, in this paper specifically in section 3 it has been explained that permutation and substitution processes are carried out at each encryption stage to increase diffusion and confusion. We have explained that for the bit-level conversion process is carried out. For permutations we convert them into one-dimensional vectors so that they are not possible to display. As for the bitXOR substitution, the bitXOR operator will convert pixels to binary form and this should be very well known.

Reviewer#3, Concern # 7: Future work in Conclusion section should be contained.

Author response:  Thank you for your advice, we updated the conclusion section marked with turquoise highlight.

Reviewer#3, Concern # 8: Please update the newly reference. Design and analysis of multiscroll memristive Hopfield neural network with adjustable memductance and application to image encryption and Design and realization of discrete memristive hyperchaotic map with application in image encryption. The authors can cite some of these literatures

Author response:  Thank you for the advice given. We updated the manuscript and add these references in section one, see turquoise highlight.

Round 2

Reviewer 2 Report

Thank you for addressing my concern. The comparison introduced in section 4.10 has improved the quality of the text.

Author Response

Thank you very much

Reviewer 3 Report

No further comment.

Minor editing of English language required.

Author Response

Thank you very much. Minor editing of English language required, have done. We include proof of tracking and clean manuscripts.
